# Human OAT1, OAT3, OAT4 and OATP1A2 Facilitate the Renal Accumulation of Ochratoxin A

**DOI:** 10.3390/pharmaceutics17111474

**Published:** 2025-11-16

**Authors:** Anish Mahadeo, Yik Pui Tsang, Angela R. Zheng, Sydney Arnzen, Acilegna G. Rodriguez, Mark S. Warren, Zsuzsanna Gáborik, Edward J. Kelly

**Affiliations:** 1Department of Pharmaceutics, School of Pharmacy, University of Washington, Seattle, WA 98195, USA; amahadeo@uw.edu (A.M.); a6245912@uw.edu (Y.P.T.); angelarz@uw.edu (A.R.Z.); sarnzen2@gmail.com (S.A.); 2BioIVT, 3010 Kenneth St., Santa Clara, CA 95054, USA; nrodriguez@bioivt.com (A.G.R.); mwarren3@bioivt.com (M.S.W.); 3SOLVO Biotechnology, Charles River Laboratories Hungary, 1117 Budapest, Hungary; zsuzsanna.gaborik@crl.com; 4Kidney Research Institute, Seattle, WA 98104, USA

**Keywords:** ochratoxin A, chronic kidney disease of unknown etiology, renal transporters, renal proximal tubule, nephrotoxicity, transporter-mediated disposition

## Abstract

**Background/Objectives**: Ochratoxin A (OTA) is a widespread foodborne mycotoxin linked to chronic kidney disease of unknown etiology. Despite evidence from animal models showing OTA accumulation in the kidney, the molecular mechanisms underlying its renal disposition in humans remain only partially understood. Here, we identify human renal transporters responsible for OTA kidney accumulation, elimination, and establish Michaelis–Menten kinetics under matched conditions to directly compare transport mechanisms. We also aim to identify inhibition potential of these transport mechanisms with common dietary polyphenols. **Methods**: Mammalian cells and membrane vesicles overexpressing human renal transporters were used to screen and profile the uptake and efflux of OTA. Miquelianin, (-)-Epicatechin-3-O-gallate, myricetin, luteolin, and caffeic acid were tested as potential concentration-dependent transporter inhibitors. **Results**: We demonstrate that OTA is a substrate for human organic anion transporter (hOAT) 1 (K_m_: 2.10 ± 0.50 μM, V_max_: 396.9 ± 27.0 pmol/mg/min), hOAT3 (K_m_: 2.58 ± 0.83 μM, V_max_: 141.4 ± 30.3 pmol/mg/min), hOAT4 (K_m_: 6.38 ± 1.45 μM, V_max_: 96.9 ± 18.8 pmol/mg/min), and human organic anion transporting polypeptide (hOATP) 1A2 (K_m_: 37.3 ± 6.2 μM, V_max_: 801.0 ± 133.9 pmol/mg/min). Among efflux transporters, OTA was transported only by human breast cancer resistance protein (hBCRP), which has minimal renal expression. While none of the uptake transporters were potently inhibited (>90%) by polyphenols at 10 μM, luteolin inhibited hBCRP-mediated transport of OTA with an IC_50_ of 22 μM and caffeic acid stimulated hBCRP-mediated efflux with an EC_50_ of 713.8 μM, both of which are physiologically relevant intestinal lumen concentrations. **Conclusions**: Our results confirm that exposure to OTA will lead to renal accumulation and increased health risks in affected populations, necessitating increased scrutiny of our food sources.

## 1. Introduction

Ochratoxin A (OTA) is a mycotoxin produced by several species of the Aspergillus and Penicillium genera of fungi and is the most common food contaminant found in numerous agricultural products, such as coffee, grains, wine, dried fruits and spices, with high stability under food-processing conditions [1]. One study found that in the United States, 52% of cereal products were contaminated with OTA [2]. OTA is a nephrotoxin, hepatotoxin, and has recently also been investigated as a neurotoxin [3,4,5]. With respect to human health, OTA has been hypothesized to be a causative agent in a type of endemic, environmentally based chronic kidney disease termed chronic kidney disease of unknown etiology (CKDu) in regions such as Sri Lanka and Tunisia, where it was found that 93.5% of chronic kidney disease patients in the North Central Province of Sri Lanka had detectable levels of ochratoxins in their serum [6,7]. Kidney biopsy and ultrastructural studies of CKDu indicate a tubulointerstitial process with predominant proximal tubular injury, with minimal primary glomerular involvement [8,9]. This makes proximal-tubule transport particularly relevant to OTA nephrotoxicity. While comprehensive toxicokinetic data for OTA in humans are limited due to ethical constraints, concerningly, one study reported an oral half-life of 35.5 days in humans, highlighting the potential for prolonged systemic exposure and sustained toxicity risk [10].

Following absorption, OTA accumulates in the kidney in animal models where it exerts nephrotoxic effects [11,12]. Schwerdt et al. showed that a 1.25 mg/kg OTA intraperitoneal dose to Wistar rats resulted in a 12-fold higher OTA concentration in the kidney compared to the unbound concentration in the plasma [11]. Similarly, Zepnik et al. found that after a 0.5 mg/kg oral dose administered to rats, OTA had accumulated to over 400 pmol/g of kidney tissue after 24 h, 12-fold that of the liver. They concluded that with high kidney concentrations and slow elimination (15% of the dose recovered in urine and feces after 96 h), OTA may undergo extensive renal reabsorption [12]. Although evidence from animal studies indicates accumulation of OTA in the kidney, potential involvement of transporter-mediated mechanisms of OTA excretion and reabsorption are not well understood in humans.

Renal proximal tubular epithelial cells (PTECs) are the primary sites of xenobiotic active secretion into the urinary filtrate. These cells express a spectrum of membrane transporters including organic anion transporters (OATs), organic cation transporters (OCTs), and ATP-binding cassette (ABC) transporters. There have been reports in rodents showing the capability of some of these transporters to facilitate the renal uptake and efflux of OTA. In mice expressing human OAT1 and OAT3, these transporters were found to facilitate high–affinity uptake of OTA into the mouse proximal tubules [13,14]. In mouse proximal tubule cells transfected with human OAT4, a saturable and inhibitable increase in OTA uptake was observed compared with mock-transfected cells [15]. Furthermore, in Caco-2 cells, a cell line derived from human colorectal adenocarcinoma, Schrickx et al. demonstrated that OTA intracellular concentrations increased in the presence of Multidrug Resistance-associated Protein 2 (MRP2) and Breast Cancer Resistance Protein (BCRP) inhibitors, suggesting OTA to be a substrate for both [16]. However, it is important to note that orthologs for SLC transporters between species have different substrate specificities and kinetic parameters. Animals and transgenic animals expressing human transporters have been reported to lack the expected pharmacokinetic behavior of certain drugs. For instance, Salphati et al. demonstrated little difference between rosuvastatin profiles in human organic anion transporting polypeptide (OATP)1B1-humanized mice compared to Oatp1a/1b knockout mice [17]. Thus, animal-based data may not always translate to or reflect human susceptibility to toxins, and more studies are needed to elucidate mechanisms of OTA transport in the human kidney.

In this study, we aim to address this knowledge gap by systematically profiling the role of major human renal transporters involved in OTA uptake, secretion and reabsorption. Using mammalian cell lines (i.e., Human Embryonic Kidney 293 (HEK293) cells, Madin-Darby Canine Kidney II (MDCK-II) cells, and Chinese Hamster Ovary (CHO) cells) and Spodoptera Frugiperda 9 (Sf9) cell-derived membrane vesicles overexpressing the major human renal transporters, we characterized OTA transport kinetics that are translatable to the human proximal tubule. This work builds upon previous studies to identify renal transporters not previously implicated in animal models in the disposition of OTA. In addition, we investigated the potential of several natural antioxidants to inhibit OTA transport. By elucidating specific mechanisms for OTA renal secretion and accumulation, this work aims to shed light on the mechanistic toxicology of OTA and the risk of exposure to kidney health in the context of environmental kidney disease.

## 2. Materials and Methods

### 2.1. Chemicals and Reagents

Ochratoxin A and ochratoxin A-^13^C_20_ were purchased from Cayman Chemical (Ann Arbor, MI, USA). [^3^H]ochratoxin A (10 Ci/mmol, 100 μM) was purchased from American Radiolabeled Chemicals (St. Louis, MO, USA). High Glucose Dulbecco’s modified Eagle’s medium (DMEM, supplemented with 4.5 g/L D-glucose, 2 mM L-glutamine, 110 mg/L sodium pyruvate), Low-Glucose DMEM (supplemented with 1 g/L D-glucose, 2 mM L-glutamine, 110 mg/L sodium pyruvate), Penicillin-Streptomycin (10,000 U/mL), Dulbecco’s phosphate-buffered saline (DPBS), Hank’s balanced salt solution with calcium and magnesium (HBSS++), heat-inactivated fetal bovine serum (FBS), GlutaMAX, Pierce Bicinchoninic Acid (BCA) Protein Assay Kit, 96-well microplates, poly-D-lysine (0.1 mg/mL), sodium hydroxide (NaOH) pellets, 37% hydrochloric acid (HCl), Optima acetonitrile (Liquid chromatography-tandem mass spectrometry (LC/MS) grade), Optima water (LC/MS grade), Optima methanol (LC/MS grade), formic acid (LC/MS grade), sodium hydroxide, and hydrochloric acid were purchased from Thermo Fisher Scientific (Waltham, MA, USA). Probenecid, bromsulphthalein, pyrimethamine, ergothioneine, verapamil, (-)-epicatechin gallate, caffeic acid, luteolin, miquelianin, myricetin, GlySar, estradiol glucuronide, prazosin, and quinidine were purchased from MedChemExpress (Princeton, NJ, USA). Ultima Gold scintillation fluid and flexible insert plates for measurement of radioactivity were purchased from Revvity (Shelton, CT, USA). Rat tail type I collagen was obtained from BioMedTech Laboratories (Tampa, FL, USA).

### 2.2. Cell Culture

HEK293 cells stably transfected with the mock vector, human OAT1, OAT2, OAT3, OAT4, OCT2, organic cation/carnitine transporter 1 (OCTN1), OATP1A2, multidrug and toxin extrusion proteins (MATE) 1 and 2-K, and CHO cells stably transfected with the mock vector and OCTN2 were generously provided by SOLVO Biotechnology (Budapest, Hungary). HEK293 cells transfected with OATP1A2 were generously provided by Dr. Jashvant Unadkat at the University of Washington. Cells were cultured in high glucose (HEK293 cells) or low-glucose (CHO cells) DMEM medium supplemented with 10% heat-inactivated FBS, 2 mM L-glutamine, 2 mM GlutaMAX, 100 IU/mL penicillin, and 100 µg/mL streptomycin. Cell culture plasticware was coated with 0.05 mg/mL of poly-D-lysine in 10 mL of DPBS (for T75 flasks) to promote cell attachment. MDCK-II cells were kindly provided by BioIVT (Santa Clara, CA, USA) and were cultured in DMEM with low glucose, low sodium bicarbonate, and 10% heat inactivated FBS. All cell lines were grown in a 37 °C incubator with 5% carbon dioxide in air and 95% humidity.

### 2.3. Uptake Assays in Transporter-Overexpressing Mock HEK293 and CHO Cells

The following experiments were conducted at the University of Washington. Mock HEK293 cells and transporter-overexpressing HEK293 cells (i.e., OAT1–4, OCT2, MATE1/2K, OCTN1, OATP1A2), mock CHO cells, and OCTN2-overexpressing CHO cells were plated in 96 well plates coated with 0.05 mg/mL of poly-D-lysine in 50 μL DPBS at a density of 800,000 cells/mL and grown until reaching over 90% of confluence. Activity of transporters in these cell lines was verified in a prior study [18]. Uptake assays were carried out in triplicate across three independent experiments at 37 °C in HBSS++ containing 10 μM OTA with or without the corresponding prototypical transporter inhibitors (i.e., 200 μM probenecid for OAT1/3, 200 μM bromsulphthalein for OAT2/4 and OATP1A2, 200 μM pyrimethamine for OCT2 and MATE1/2-K, 1 mM ergothioneine for OCTN1, 200 μM verapamil for OCTN2). For uptake experiments in MATE1- and MATE2-K-expressing HEK293 cells, the pH of the incubation buffer was adjusted to 8.0 to facilitate substrate uptake. For uptake experiments in OATP1A2-expressing and OAT4-expressing HEK293 cells, the pH of the incubation buffer was adjusted to 6.5 to mimic pH of the urinary filtrate, as these are apical transporters. The incubation buffer was kept at pH 7.4 for all other transporters. After three warm washes of cells with HBSS++, substrates, with or without inhibitors, were incubated for 5 min (confirmed to be within the linear uptake range). Uptake was quenched by removing the substrates/inhibitors and washing the cells 3 times with 200 μL of ice-cold HBSS++. Cells were then lysed in 100 μL acetonitrile with 100 nM internal standard (i.e., OTA-^13^C_20_). Without disturbing the precipitated layer of proteins, 75 μL of samples were diluted 1:1 with water before OTA quantification using LC-MS/MS. The precipitated protein was allowed to dry and subsequently solubilized overnight in 1M NaOH, neutralized with 1 M HCl the next day, and measured by BCA assay. Kinetic parameters for identified OTA transporters were determined under the same conditions using concentration-dependent uptake assays in triplicate across three independent experiments.

### 2.4. Uptake Assays in MRP2 and MRP4-Expressing Sf9 Membrane Vesicles

The following experiments were conducted at BioIVT. Transport assays using Sf9 membrane vesicles were described previously [18]. Briefly, control membrane vesicles and those expressing either human MRP2 or MRP4 (BioIVT) were conducted in vesicle uptake buffer (50 mM MOPS-Tris pH 7.0, 70 mM KCl, 7 mM MgCl_2_) with 3 mM glutathione added as the reducing agent. Activity of MRP2 and MRP4 in these vesicles was verified using probe substrates in all experiments (Appendix A) and in a previous study [18]. Vesicles were first pre-mixed with 10 μM OTA (total concentration of both unlabeled OTA and 10 nM [^3^H] OTA corresponding to a 10 Ci/mmol stock), and aliquots of the vesicle-substrate mixture were added as a suspension onto 96-well flat bottom assay plates before a 10 min preincubation at 37 °C with orbital shaking at 60 rpm. Uptake was initiated by adding either AMP or ATP at a final concentration of 5 mM. Plates were incubated at 37 °C with 60 rpm orbital shaking for 5 min (confirmed to be within the linear uptake range). Transport was terminated by adding ice-cold vesicle wash buffer (40 mM MOPS-Tris pH 7.0, 70 mM KCl), and the mixture was transferred to a 96-well glass fiber filtration plate. Vacuum was then applied, and the vesicles were washed 3 times with ice-cold vesicle wash buffer. Vesicles were then lysed with 50% ACN in water and mixed with Ultima Gold scintillation fluid. The radioactivity of each sample was measured by Wallac 1450 Microbeta liquid scintillation counter (Perkin Elmer, Shelton, CT, USA). Each experimental condition was performed in triplicate across 3 independent experiments. Total protein content of vesicles was measured by BCA assay.

### 2.5. Uptake Assays in PEPT1, PEPT2, and URAT1-Expressing MDCK-II Cells

Transport assays using MDCK-II cells were described previously [18]. Briefly, MDCK-II cells expressing PEPT1, PEPT2, and URAT1 (BioIVT) were seeded at a density of 60,000 cells per well on 96-well polycarbonate transwell membrane inserts (0.4 μm pore size) coated with rat tail type I collagen. Approximately 24 h after seeding, confluent MDCK-II cell monolayers were transfected with DNA plasmids encoding either PEPT1, PEPT2, URAT1, or green fluorescent protein as a negative control (mock cells) at a final concentration of 30 ng/μL for 48 h using Opti-Expression, a novel in situ transfection technology that ensures effective transporter transfection in polarized cell monolayers. The activity of these transporters was confirmed using probe substrates (Appendix A). All experiments were conducted in triplicate across 3 independent experiments. The pH of the substrate incubation buffer was adjusted to 6.5 to mimic pH of the urinary filtrate, as these transporters are expressed apically in vivo. Cells were first washed 3 times with warm HBSS++ and preincubated with HBSS++ in the apical compartments for 30 min at 37 °C. Substrate uptake was initiated by replacing the preincubation buffer with buffer containing 10 μM OTA (total concentration of both unlabeled OTA and 10 nM [^3^H] OTA corresponding to a 10 Ci/mmol stock), or the vehicle control (i.e., 0.1% ethanol) in the apical compartment at 37 °C. After 5 min of incubation (confirmed to be within linear range), cells were washed 4 times with ice-cold DPBS and lysed with 50% ACN in water. Cell lysates were mixed with Ultima Gold scintillation fluid, and the radioactivity of each sample was measured by Wallac 1450 Microbeta liquid scintillation counter (Perkin Elmer). Each experimental condition was performed in triplicate across 3 independent experiments. Total protein content of cells was measured by BCA assay.

### 2.6. Bidirectional Transport Assays in BCRP and P-gp-Expressing MDCK-II Cells

For BCRP, MDCK-II cells were seeded at a density of 60,000 cells per well on 96-well polycarbonate transwell membrane inserts (0.4 μm pore size) coated with rat tail type I collagen. Approximately 24 h after seeding, fully confluent MDCK-II cell monolayers were transfected with DNA plasmids encoding either BCRP or green fluorescent protein as a negative control (mock cells) at a final concentration of 30 ng/mL using Opti-Expression. The cells were incubated for approximately 48 h under normal culture conditions (37 °C, 5% carbon dioxide in air, and 95% humidity) to allow BCRP to be expressed and localized on apical membranes. For P-gp, MDCK-MDR1 cells (or MDCK-II parental cells as a negative control) were seeded at a density of 60,000 cells per well on 96-well polycarbonate transwell membrane inserts (0.4 μm pore size) coated with rat tail type I collagen. The cells were incubated for approximately 72 h under normal culture conditions to allow cell monolayers to become fully confluent and P-gp to be expressed and localized on apical membranes. Transporter activity of both BCRP and P-gp in these cells was verified using probe substrates in all experiments (Appendix A). Prior to the transport experiments, cells were washed 3 times with warm HBSS++ to remove the culture media and preincubated in HBSS (containing inhibitor, if applicable, or vehicle control if no inhibitor was used) in both the apical and basolateral compartments for 30 min at 37 °C. The pH of the incubation buffer in the apical chamber was adjusted to 6.5 while that in the basal chamber was kept at 7.4 to mimic the microphysiological environment of PTECs in vivo. Substrate transport was initiated by replacing the preincubation buffer with buffer containing 10 μM OTA (total concentration of both unlabeled OTA and 10 nM [^3^H] OTA corresponding to a 10 Ci/mmol stock) (and inhibitor, if applicable, or vehicle control) in the donor chamber and buffer (with either inhibitor, if applicable, or vehicle control) in the receiver chamber at 37 °C. The apical chamber served as the donor chamber for apical to basolateral (A to B) experiments while the basolateral chamber served as the donor chamber for basolateral to apical (B to A) experiments, with the opposite chamber being the receiver chamber. After the end of the 60 min incubation, aliquots were taken from both donor and receiver chambers and mixed with Ultima Gold scintillation fluid. The radioactivity of each sample was measured by Wallac 1450 Microbeta liquid scintillation counter (Perkin Elmer, Shelton, CT, USA). Efflux ratios were determined by dividing the B to A permeability by the A to B permeability.

### 2.7. Transporter-Mediated Natural Product-OTA Interaction Study

OTA transporters identified in the uptake screen (OAT1, OAT3, OAT4, OATP1A2, BCRP) were further evaluated for modulation by 5 diet-relevant polyphenols (ECG, miquelianin (quercetin 3-O glucuronide), caffeic acid, luteolin, and myricetin). Because OATP1A2 is expressed at exceptionally low levels in the apical membrane of the kidney as well as in the liver and intestine, interaction studies focused on OAT1, OAT3, OAT4, and BCRP as these will have a greater impact on circulating polyphenol levels. For the initial inhibitor screen, HEK 293 cells expressing OAT1, OAT3, or OAT4 were incubated for 5 min with 300 nM OTA in the absence (vehicle, 0.1% dimethyl sulfoxide) or presence of a single polyphenol (10 µM). Uptake assays were performed in three independent experiments, each in technical triplicate.

Because luminal concentrations in the intestine of these polyphenols can reach high micromolar to millimolar levels after oral intake, concentration-dependent effects on BCRP-mediated OTA efflux were characterized instead of a single-point screen. The concentrations of polyphenols tested were as follows: ECG and miquelianin: 100, 200, 400, 600, 800, 1000 μM; caffeic acid: 50, 100, 500, 1000, 5000, 10,000 μM; luteolin: 0.1, 1, 10, 20, 25, 40, 50, 60, 100, 150 μM; myricetin: 5, 10, 50, 100, 250, 500, 650, 750, 850, 1000 μM. All other assay conditions were as described above. Transport activity was expressed as percent of vehicle control and fitted to a four-parameter logistic model to obtain half-maximal inhibitory concentration (IC_50_) values (for inhibition) or half-maximal effective concentration (EC_50_) values (for activation). Three independent experiments, each with technical triplicate, were performed for every compound.

### 2.8. Quantification of OTA by LC-MS/MS

A Xevo TQ-XS triple quadrupole mass spectrometer coupled with an ACQUITY ultra-performance liquid chromatography and an electrospray ionization source (Waters, Milford, MA, USA) was used for the quantification of OTA. Detailed LC-MS/MS conditions are provided in Appendix A.

### 2.9. Data and Statistical Analysis

All cellular-based assays were conducted in technical triplicate across three independent experiments. Data acquisition and quantitative analysis of OTA were performed using Micromass MassLynx software version 4.1 (Waters). For transporter screening assays, OTA was classified as a substrate for a given transporter when its cellular accumulation or substrate flux was ≥twofold that of matched negative controls. This cutoff was defined a priori. For bidirectional transport assays for BCRP and P-gp, apparent permeability was calculated as follows:$$P_{app}=\frac{dQ}{dt}\times \frac{1}{A\times C_{0}}$$
where *P_app_* is the apparent permeability of compounds, *Q* is the amount of OTA transported over time *t, A* is the insert membrane surface area, and *C*_0_ is the initial compound concentration in the donor chamber. Efflux ratios were subsequently determined by dividing the B to A permeability by the A to B permeability:$$ER=\frac{P_{app}\;(B\;to\;A)}{P_{app}\;(A\;to\;B)}$$
where *P_app_ (B to A*) is the apparent permeability of OTA from the basolateral compartment to the apical compartment, and *P_app_ (A to B)* is the apparent permeability of compounds from apical compartment to the basolateral compartment. For transporter kinetics, *K_m_* and *V_max_* values were obtained by fitting the Michaelis–Menten equation to the substrate-concentration data,$$V=\frac{V_{max}*[S]}{K_{m}+[S]}$$
where *V* is the initial uptake rate at OTA concentration [*S*], *V_max_* is the maximum transport rate, and *K_m_* is the concentration of OTA that produces half-maximal transport. For efflux transporter kinetics determined by bidirectional transwell assays, OTA flux was calculated by multiplying P_app_ values with the initial concentration applied to the donor chambers (i.e., $J=P_{app}\times C_{0}$). The net flux (bidirectional ΔFlux, *J_net_*) was then calculated by the following equation:$$J_{net}=\left(J_{B-to-A\left(Transporter\right)}-\;J_{A-to-B\left(Transporter\right)}\right)-\left(J_{B-to-A\left(Mock\right)}-\;J_{A-to-B\left(Mock\right)}\right)$$
where *J_net_* is the bidirectional ΔFlux, *J_B-to-A(Transporter)_* is the OTA flux from basal chamber to apical chamber in transporter-expressing cells, *J_A-to-B(Transporter)_* is the OTA flux from apical chamber to basal chamber in transporter-expressing cells, *J_B-to-A(Transporter)_* is the OTA flux from basal chamber to apical chamber in mock cells, *J_A-to-B(Transporter)_* is the OTA flux from apical chamber to basal chamber in mock cells. *J_net_* was then treated as the initial uptake rate, and *K_m_* and *V_max_* values were obtained by fitting the Michaelis–Menten equation to the substrate-concentration data as described above. Intrinsic clearance (*CL_int_*) was then calculated by the formula,$${CL}_{int}=\frac{V_{max}}{K_{m}}$$
using *V_max_* and *K_m_* values as calculated above. Intracellular OTA concentration with respect to the BCRP kinetics assay were based on a three-compartment model at steady state where *A* represents the apical OTA concentration, *B* represents the basal concentration, and *C* represents the intracellular concentration. At steady-state, the mass balance for intracellular concentrations assuming symmetric passive permeabilities (*P_apical_* = *P_basal_*, taken as *P_app_* from mock assays) and apical efflux gives:$$C_{cell}=\;\frac{C_{basal}}{1+\;\frac{k_{e}}{P_{a}}}$$
where *C_basal_* is the concentration in the basal chamber and *k_e_* is the first-order elimination with respect to BCRP. Because the asymmetric efflux ratio ${ER}_{a}$ is defined as:$${ER}_{a}=1+\;\frac{k_{e}}{P_{a}}$$

Substitution into the previous equation and rearrangement to solve *C_cell_* allows for calculation of intracellular concentrations from efflux ratios and OTA concentrations:$$C_{cell}=\;\frac{C_{basolateral}}{{ER}_{a}}$$

Fraction-of-control data from polyphenolic natural products as OTA transport inhibitors were analyzed by one-way ANOVA followed by post hoc Dunnett’s multiple comparison test. A *p*-value < 0.05 was considered statistically significant. For the estimation of *IC*_50_ or *EC*_50_ values, transport activity data were fitted with the following four-parameter logistic models:$$V=Bottom+\frac{Top-Bottom}{1+{\left(\frac{\left[I\right]}{{IC}_{50}}\right)}^{h}}\;\left(for\;inhibitors\right)$$$$V=Bottom+\frac{{\left[I\right]}^{h}\times (Top-Bottom)}{{\left[I\right]}^{h}+{{EC}_{50}}^{h}}\;\left(for\;activators\right)$$
where *V* is transport activity expressed as a percentage of vehicle control, [*I*] is inhibitor (or activator) concentration, Top and Bottom are the asymptotic plateaus, *IC*_50_ or *EC*_50_ is the concentration producing half-maximal inhibition (or activation), and h is the Hill slope. All data analyses were performed using GraphPad Prism version 10.2.3 (GraphPad). Parameter estimates are reported as best-fit ± standard error from three independent experiments, each run in technical triplicate.

## 3. Results

### 3.1. Human Renal Transport Screen for OTA

Among the uptake transporters tested, OTA was shown to be a substrate for OAT1, OAT3, OAT4, and OATP1A2 with fold uptake relative to negative controls (in the presence of transporter inhibitors) of 10.7, 7.6, 3.0, and 3.3, respectively (Figure 1A). OTA was also shown to be a substrate for BCRP, with an efflux ratio relative to mock cells of 7.8 ± 1.2 (Figure 1B). OTA was not a substrate for all other tested transporters.

### 3.2. Transporter Kinetics of OTA via OAT1, OAT3, OAT4, OATP1A2, and BCRP

After confirming that OTA is transported by OAT1, OAT3, OAT4, OATP1A2, and BCRP, we quantified their kinetic parameters. For the uptake transporters (OAT1, OAT3, OAT4, OATP1A2), OTA accumulation was corrected for mock-cell uptake, normalized to cellular protein amount, and analyzed by nonlinear regression to obtain K_m_, V_max_, and intrinsic transport clearance (CL_int_ = V_max_/K_m_; Table 1). Both OAT1 and OAT3 exhibited high–affinity OTA transport, with K_m_ values of 2.10 ± 0.50 and 2.58 ± 0.83 μM, respectively, and an intrinsic clearance for OAT1 of 194.4 ± 34.4 µL/mg total protein/min and 56.1 ± 6.3 µL/mg total protein/min for OAT3 (Figure 2A,B, Table 1). The apical uptake transporter OAT4 showed comparably high affinity with a K_m_ value of 6.38 ± 1.45 μM (Figure 2C, Table 1). The distal uptake transporter OATP1A2 had lower affinity with a K_m_ value of 37.3 ± 6.2 μM (Figure 2D, Table 1). At higher concentrations of OTA (25 to 50 μM), OAT1 uptake continued to rise, whereas OAT3 uptake declined, suggesting a secondary binding site for OAT1 and saturation or substrate auto-inhibition for OAT3 (Figure 2A,B, Table 1). For BCRP, net B to A flux increased linearly from 1.56 to 100 μM OTA without evidence of saturation (Figure 2E). Therefore, individual kinetic parameters could not be estimated, and the apparent CL_int_ estimated by the slope of the linearly fitted line is 0.29 ± 0.03 µL/min/cm^2^. At the highest tested concentration (100 μM), the steady-state intracellular concentration of OTA was estimated to be 7.5 ± 0.9 μM.

### 3.3. Natural Product-OTA Interactions at Human OAT1, OAT3, and OAT4

Many natural polyphenols and dietary antioxidants are known substrates of OAT1 and OAT3, which we and others have shown to be dominant basolateral uptake transporters of OTA. Next, we assessed whether these compounds have inhibitory potential of OTA uptake. We co-incubated 300 nM of OTA (below the K_m_ values for OAT1 and OAT3) with five polyphenols reported to interact with renal transporters [19,20,21]: ECG, miquelianin (quercetin 3-O glucuronide), caffeic acid, luteolin (each at 10 μM). OTA uptake in the presence of these polyphenols in transporter-expressing cells was normalized to that in mock cells and is expressed as a fraction of vehicle control (OTA only) in Figure 3. Luteolin was the most potent inhibitor, significantly reducing OTA uptake via OAT1, OAT3, and OAT4 to 0.28, 0.36, and 0.7-fold of vehicle control, respectively. Miquelianin significantly inhibited OAT3 (0.58-fold of control) but had no significant effects on OAT1 or OAT4. Myricetin caused a modest but significant reduction in OTA uptake through all three transporters (fraction of vehicle control: 0.73 for OAT1, 0.74 for OAT3, 0.82 for OAT4). ECG and caffeic acid did not significantly inhibit OTA transport under these conditions.

### 3.4. Natural Product-OTA Interactions at Human BCRP

Because OTA was also transported by BCRP and that BCRP is highly expressed in other organs (e.g., liver, intestine, blood–brain barrier), we characterized the same 5 polyphenols for their ability to modulate OTA extracellular efflux. OTA transport was measured in MDCK-II cells expressing human BCRP at a fixed substrate concentration of 300 nM, while each polyphenol was tested over different concentrations within the 1 to 10,000 μM range where solubility allows. Activity is expressed as a percentage of vehicle control (100%). As shown in Figure 4, luteolin appeared to be the most potent inhibitor of BCRP-mediated OTA transport among the 5 polyphenols, with an IC_50_ value of 22.0 μM. ECG and myricetin were weak inhibitors with IC_50_ values of 495.1 μM and 424.8 μM, respectively. Interestingly, caffeic acid appeared to stimulate the activity of BCRP, with an EC_50_ value of 713.8 μM. Miquelianin showed no appreciable inhibition or activation.

## 4. Discussion

In this study, we systematically characterized OTA transport by key human renal transporters using heterologous expression systems with minimal endogenous transporter activity. Rather than solely invoking direct cytotoxicity of OTA, our data show that transporter-mediated basolateral uptake (OAT1/3) and apical reabsorption (OAT4, OATP1A2) can drive renal accumulation of OTA, notably in the proximal tubule, which provides a mechanistic basis for how chronic exposure to OTA could contribute to kidney injury, including CKDu. These findings are consistent with CKDu biopsy and ultrastructural studies that demonstrate a tubulointerstitial pattern with predominant proximal tubular injury, supporting our identification of OAT1/3 and OAT4 as key determinants of OTA accumulation [8,9].

We confirm prior reports that OTA is avidly taken up by basolateral OAT1 and OAT3 in PTECs [14]. Given OTA’s long half-life and extensive albumin binding in human plasma, filtration of the unbound fraction can account for much of the very low total systemic clearance at dietary-level exposures [10,12,22]. Our data nonetheless provide a mechanistic basis for active tubular handling of OTA. OAT1/3 mediate basolateral uptake of OTA with low-μM apparent K_m_ values (Figure 1 and Figure 2, Table 1), consistent with prior studies conducted in animals or humanized models [14]. At typical circulating total concentrations of OTA (~2 nM in the general population, up to 1.6 μM in CKDu patients [6,7]), renal transport of unbound OTA is in the linear range (C << K_m_). Consistent with this mechanism, prior report showed that combined OAT1/3 inhibition by estrone sulfate and para-aminohippurate nearly abolished OTA uptake in intact rabbit proximal tubules [23]. These data support that renal elimination of OTA at physiologically relevant concentrations involves both filtration of the unbound fraction and OAT1/3-mediated transport.

In addition to basolateral uptake by OAT1/3, our data indicate that OAT4 on the apical membrane of the proximal tubule also plays a key role in OTA disposition. We observed high–affinity OTA uptake by OAT4 (K_m_ = 6.38 ± 1.45 μM) (Figure 1 and Figure 2, Table 1), an affinity similar to that reported previously for hOAT4 expressed in Chinese hamster ovary (CHO) cells [24]. OAT4 can function as a bidirectional organic anion/dicarboxylate exchanger, and the high concentrations of intracellular dicarboxylates (e.g., succinate) in proximal tubule cells drive OAT4 to uptake anions like OTA from the tubular lumen. Thus, our data suggests that OAT4-mediated OTA transport contributes to its accumulation in the kidney. Additionally, it is important to note that the transport assays for OAT4 were conducted at pH 6.4 to simulate the acidic urinary filtrate, as OAT4 has been shown to utilize OH^−^ in urate reabsorption, and enhanced resorptive activity was observed upon acidification of the extracellular medium [25]. Consequently, when urine is more acidic, OAT4-mediated OTA reabsorption is expected to increase, which further contributes to OTA retention in the kidney.

We also evaluated OATP1A2 as an OTA transporter. OATP1A2 is one of the most important uptake transporters in the blood–brain barrier that has a broad substrate specificity, and it has been shown to be expressed on the apical membrane of distal tubule cells by Northern blot and immunohistochemical staining analysis [26,27]. Here, we provide kinetic parameters of OTA reuptake by OATP1A2 (K_m_ = 37.3 ± 6.2 μM) (Figure 1 and Figure 2, Table 1), which suggests a role of kidney distal tubule reabsorption of OTA. OATP1A2-mediated uptake has been shown to be strongly pH-dependent. The rate of methotrexate uptake mediated by OATP1A2 has been reported to be 5- to 7-fold higher at pH 5.0 compared to pH 7.4 [27]. While our uptake experiments were conducted at pH 6.5, these results suggest that more acidic urine would further enhance OTA reabsorption by OATP1A2. In the context of CKDu, this is an important insight. In Central America, heat-stress induced dehydration, leading to urinary volume depletion and acidification, has been associated with Mesoamerican nephropathy [28]. However, multivariate regression analysis of CKDu risk factors present in Sri Lanka did not show a significant effect of heat stress and dehydration [29]. Our data suggests that such conditions could markedly increase OTA reabsorption through OATP1A2 (and OAT4), thereby increasing the retention of OTA and possibly other nephrotoxins in the kidney. This mechanistic link provides a plausible explanation for why heat stress and dehydration might predispose certain populations (e.g., agricultural workers) to OTA-related kidney injury.

In addition to uptake mechanisms mentioned above, we tested a broad panel of other renal uptake and efflux transporters and found that only BCRP measurably transported OTA (Figure 1). Earlier work by Schwerdt et al. implicated PEPTs in the tubular reabsorption of OTA [30,31]. However, we did not observe any hPEPT-mediated transport of OTA (Figure 1). Because those earlier studies used MDCK cells, it is possible that canine PEPT isoforms recognize OTA as a substrate whereas their human orthologs do not. In addition, MRP2 has also been suggested to transport OTA. For instance, OTA was found to inhibit MRP2-mediated p-aminohippurate efflux with an IC_50_ of 58 μM, and coadministration of OTA with the MRP2 inhibitor MK-571 or the BCRP inhibitor Ko143 increased intracellular OTA in Caco-2 cell monolayers, suggesting that both transporters may contribute to OTA efflux [16]. Recently, Qi et al. reported modest MRP2-mediated transport of OTA at pH 6.4, although significant differences in A-to-B versus B-to-A flux emerged only after 8 h of incubation and no inhibitor control was included [32]. In contrast, we observe no MRP2-mediated OTA transport, whereas BCRP showed robust and strictly linear OTA B→A flux up to a calculated intracellular concentration of 7.5 ± 0.9 μM without evidence of saturation (Figure 1B and Figure 2E), indicating that BCRP has a high capacity, but low affinity, for OTA transport. While monolayer efflux assays do not directly assess true efflux kinetics, the lack of saturation at this concentration of intracellular OTA indicates the true K_m_ is much higher, beyond a physiologically relevant exposure. The true K_m_ value of BCRP transport of OTA will require future experiments in other systems, such as BCRP-expressing membrane vesicles. Despite this observation, the contribution of renal BCRP in vivo will likely be minimal due to its low mRNA expression in the proximal tubule as well as low protein abundance in human cortex samples (below limits of detection) as measured by targeted proteomics [33,34,35]. Collectively, our findings indicate that the renal transporter-mediated efflux of OTA is minor relative to its pronounced uptake and further reinforce the view that OTA accumulation in the kidney is driven by renal uptake processes.

This work presents a novel assessment of whether common antioxidant polyphenols, found in leafy greens, teas, and coffee, could reduce renal uptake and therefore exposure by competitively inhibiting OAT1, OAT3, OAT4, or BCRP, as these foodstuffs are common dietary sources of OTA as well. We focused on luteolin, ECG, myricetin, caffeic acid, and miquelianin, all of which are known substrates for these transporters and are sold as dietary supplements. These compounds have also been shown, both in vitro and in vivo, to reduce OTA toxicity through direct free-radical scavenging and by activating cytoprotective pathways such as NRF2 and HIF1α [36,37,38,39]. We observed significant but modest inhibition by ECG, miquelianin, and myricetin of OAT1, OAT3, and OAT4 (Figure 3) at 10 µM. Luteolin, however, was more potent, decreasing OAT1- and OAT3-mediated OTA uptake by 72% and 64%, respectively. However, the in vivo relevance of this effect is uncertain as the oral bioavailability of most polyphenols is low (≤10%) [40,41,42]. An unbound concentration of 10 μM of these compounds in human plasma or kidney is unlikely to be achieved through normal diet or supplementation. On the other hand, if these polyphenols are present in the gut, they could inhibit intestinal BCRP and thereby increase OTA absorption. The U.S. Food and Drug Administration (FDA) transporter drug–drug interaction guidance estimates intestinal luminal drug concentrations by dividing dose by 250 mL [43], yielding polyphenol concentrations in the gut from low μM (myricetin, luteolin, ECG) to mM (caffeic acid) at their common doses. Our data showed that luteolin, ECG, and myricetin are inhibitors of BCRP-mediated OTA transport with IC_50_ values of 22.0 μM, 495.1 μM, and 424.8 μM, respectively (Figure 4A,D,E), suggesting that they could increase systemic OTA exposure by blocking intestinal efflux.

Interestingly, caffeic acid was found to stimulate, rather than inhibit, BCRP-mediated OTA efflux, with an estimated EC_50_ value of 713.8 μM and a maximal stimulation at around 2-fold of control (Figure 4C). Because caffeic acid has been shown to allosterically stimulate the ATPase activity of P-gp [44], a similar allosteric effect on BCRP could plausibly underlie the observed increase in OTA transport. Regarding clinical relevance, evidence for human oral bioavailability of OTA is limited. The frequently cited “~93%” does not represent absolute bioavailability. This value was derived from a single fasted volunteer given a microdose of radioactive OTA in pure ethanol, and reflects the percentage of administered radioactivity detected in plasma within 8 h [10]. No intravenous dose was given to compute the absolute oral bioavailability of unchanged OTA. Moreover, those ethanol/fasted conditions likely overestimate bioavailability relative to OTA ingestion as a food contaminant. Animal studies report moderate oral bioavailability (~30 to 60%) with notable effects of age, sex, and feeding state [22,45]. Since there are significant differences in drug/toxin bioavailability, further studies in humans are needed to determine whether this activation meaningfully alters OTA disposition in humans. Thus, the net in vivo impact of caffeic acid-mediated BCRP stimulation on human OTA exposure remains uncertain. Definitive assessment will require human studies that measure unchanged OTA under fed conditions and, ideally, include an intravenous to oral dose comparison to estimate bioavailability.

Our study has several limitations. First, although OTA is >99% bound to human serum albumin (HSA), all uptake and kinetic experiments were conducted in protein-free buffers. Recent work with renal OAT1/3 showed that omitting HSA can underestimate the effective unbound CL_int_ of highly bound substrates [46]. However, this does not change the conclusion that OAT1/3 mediate high–affinity basolateral uptake of OTA. Future studies will incorporate HSA to better estimate the effective unbound CL_int_. Second, some cell lines were unavailable for several renal transporters of interest, most notably OAT10 and OATP4C1, which have also been shown to transport organic anions and uremic toxins [47,48]. Jutabha et al. have also shown that SLC17A3 (NPT4) is capable of OTA extracellular efflux, albeit with a K_m_ (802.8 µM) value much greater than those of OAT1/3/4-mediated OTA transport (Figure 2) [49]. Thus, at the sub- to low-micromolar concentrations used here and expected in vivo, NPT4-mediated efflux is inefficient and may constrain net secretion of OTA, relative to high-affinity basolateral uptake by OAT1/3 and apical reabsorption via OAT4. Nonetheless, the authors note that other SLC17 family transporters in the kidney, notably NPT1, share similar substrate specificity for organic anion transport [49]. Therefore, the role of SLC17s in OTA efflux should be further explored in future studies. Third, in rats, low doses of OTA have been shown to upregulate rat Oats and lead to increased kidney accumulation, whereas high doses have the opposite effect [50]. Whether a similar regulatory mechanism exists in humans remains unknown and warrants investigation under chronic OTA exposure conditions. Lastly, high level of OTA binding to albumin would likely contribute to the prolonged renal retention of OTA. Future studies should measure the intracellular protein binding of OTA for better prediction of OTA accumulation in PTEC and its contribution to overall nephrotoxicity.

In conclusion, our comprehensive human renal transporter screen and kinetic analysis show that OTA renal disposition is dominated by high–affinity uptake via OAT1/3 and apical reabsorption through OAT4 and OATP1A2. BCRP-mediated efflux was first-order and non-saturable under tested OTA concentrations, but its low renal abundance suggests that this pathway likely plays a minor role in the renal clearance of OTA. Luteolin, ECG, and myricetin were found to inhibit intestinal BCRP at relevant concentrations and may increase systemic exposure of OTA, whereas caffeic acid activates BCRP and may lower absorption. Our findings provide mechanistic insights into OTA accumulation in the human kidney and its putative link to CKDu. Finally, given the ubiquitous nature of OTA in common foodstuffs and a lack of testing in the USA [51], our findings highlight the importance of regulatory guidelines for OTA in line with what other countries have implemented.

## Figures and Tables

**Figure 1 pharmaceutics-17-01474-f001:**
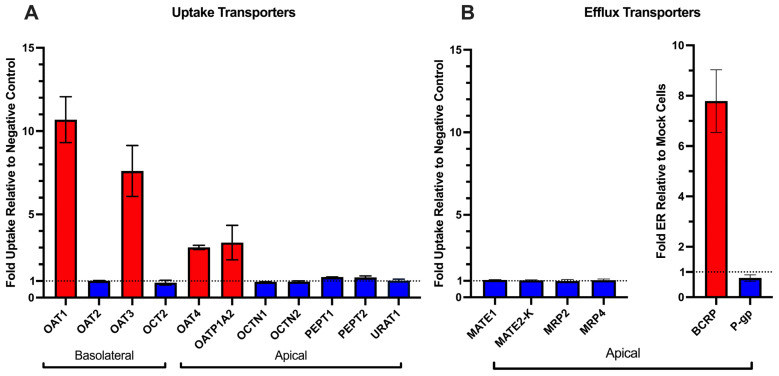
Identification of human renal OTA transporters. (**A**) uptake transporters; (**B**) efflux transporters. Data are shown as the average fold uptake or efflux ratio of 10 μM OTA ± S.D. of three independent experiments (each in technical triplicate) in individual transporter-overexpressing HEK293 cells (OAT1–4, OATP1A2, OCT2, OCTN1, MATE1, MATE2-K), CHO cells (OCTN2), MDCK-II cells (PEPT1, PEPT2, BCRP, P-gp), and Sf9 membrane vesicles (MRP2, MRP4) relative to negative controls. For HEK293 and CHO cells, prototypical transporter inhibitors were used as negative controls (i.e., 200 μM probenecid for OAT1/3, 200 μM bromsulphthalein for OAT2/4 and OATP1A2, 200 μM pyrimethamine for OCT2 and MATE1/2-K, 1 mM ergothioneine for OCTN1, 200 μM verapamil for OCTN2). For membrane vesicles, mock transfected vesicles were used as negative controls. For MDCK-II cells, transient transfection of green fluorescent protein was used as the negative control. The dashed line indicates a fold ratio of 1. Bars in red indicate the identified OTA transporter based on the a priori criteria of average fold activity of ≥2 relative to negative control, bars in blue do not meet this criteria. Transport assay duration (5 min) was confirmed to be within the linear uptake range.

**Figure 2 pharmaceutics-17-01474-f002:**
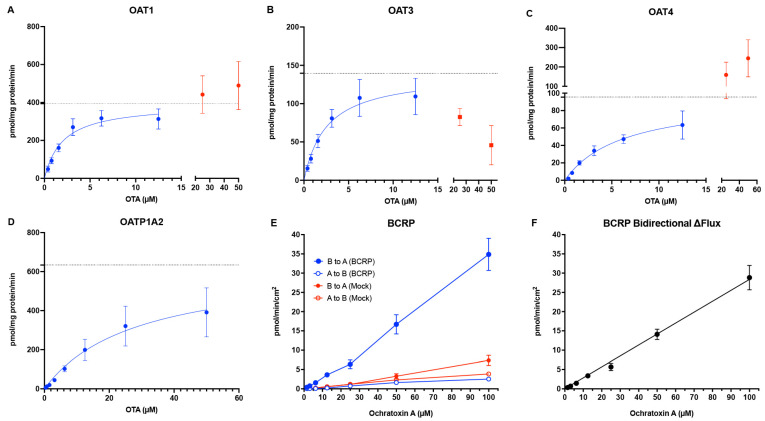
OTA transport kinetics by (**A**) OAT1, (**B**) OAT3, (**C**) OAT4, (**D**) OATP1A2, and (**E**,**F**) BCRP. Concentration-dependent transport of OTA was measured in both transporter-expressing and mock HEK293 (OAT1, OAT3, OAT4, OATP1A2) and MDCK-II cells (BCRP) after a 5 min incubation at 37 °C. For uptake transporters in HEK293 cells (**A**–**D**), transport-specific uptake was calculated by subtracting the uptake in mock cells from that in transporter-expressing cells. For BCRP kinetics, the net bidirectional flux was calculated by first subtracting the OTA flux in both BCRP-expressing cells and mock cells from the apical to basal (A to B) chamber from the OTA flux from the basal to apical (B to A) chamber (**E**). The net bidirectional flux (**F**) was then calculated by taking the difference of net B to A secretory flux in both BCRP-expressing cells and mock cells. Data were fitted with the Michaelis–Menten equation using nonlinear regression. Data shown are from 3 independent experiments with 3 technical replicates each, with the mean ± S.D. plotted. Horizontal dashed lines represent V_max_ values. Data points in red in (**A**–**C**) were not included in transporter kinetic analysis.

**Figure 3 pharmaceutics-17-01474-f003:**
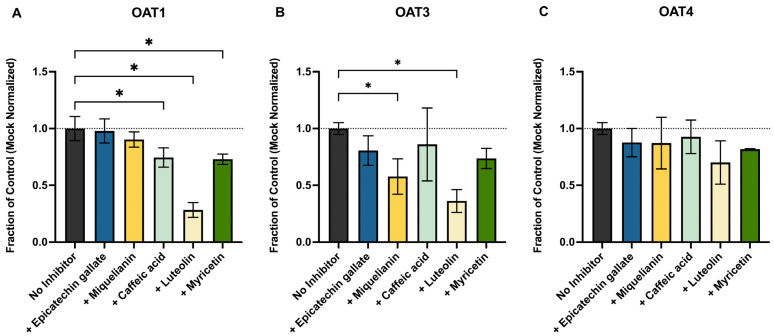
Inhibitory effect of natural antioxidants on the uptake of OTA by (**A**) OAT1, (**B**) OAT3, and (**C**) OAT4. Each inhibitor at 10 μM was incubated with 300 nM OTA for 5 min at 37 °C. OTA uptake in the absence and presence of inhibitors was measured in both transporter-expressing cells and mock cells. Transporter-specific uptake was calculated by subtracting the OTA accumulation in mock cells from that in transporter-expressing cells. The dashed line represents control transporter activity (100%). Data are shown as mean fraction of OTA uptake without inhibitors ± S.D. from 3 independent experiments, each with 3 technical replicates. Uptake in the presence of inhibitors was compared with that in the absence of inhibitors using one-way ANOVA with Dunnett’s correction (* *p* < 0.05).

**Figure 4 pharmaceutics-17-01474-f004:**
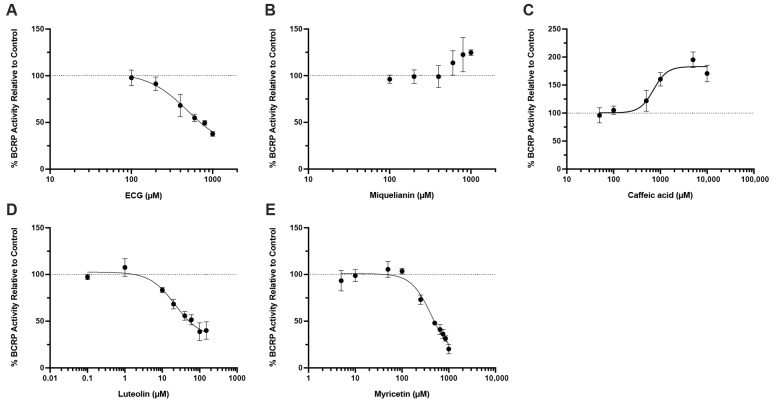
Inhibitory effect of natural antioxidants on OTA transport by BCRP. OTA (300 nM) efflux ratios in the absence and presence of (**A**) ECG, (**B**) miquelianin, (**C**) caffeic acid, (**D**) luteolin, and (**E**) myricetin were measured in both BCRP-expressing and mock MDCK-II cells. Transport of OTA by BCRP was expressed as a percentage of the efflux ratio (mean ± S.D.) in the absence of the corresponding inhibitor (vehicle control, DMSO). Data were fitted with four-parameter logistic models for inhibition (**A**,**B**,**D**,**E**) and stimulation (**C**). The dashed line represents control BCRP activity (100%).

**Table 1 pharmaceutics-17-01474-t001:** Michaelis–Menten kinetic parameters of OTA transport by OAT1, OAT3, OAT4, OATP1A2. Values are mean ± S.D. from three independent experiments.

	OAT1	OAT3	OAT4	OATP1A2
K_m_ (μM)	2.10 ± 0.50	2.58 ± 0.83	6.38 ± 1.45	37.3 ± 6.2
V_max_ (pmol/mg/min)	396.9 ± 27.0	141.4 ± 30.3	96.9 ± 18.8	801.0 ± 133.9
CL_int_ (μL/mg/min)	194.4 ± 34.4	56.1 ± 6.3	15.3 ± 0.8	21.6 ± 2.3

## Data Availability

The raw data supporting the conclusions of this article will be made available by the authors on request.

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
