# Peer review of "Human OAT1, OAT3, OAT4 and OATP1A2 Facilitate the Renal Accumulation of Ochratoxin A"

_pharmaceutics, 2025, doi:10.3390/pharmaceutics17111474_

Round 1

Reviewer 1 Report

Comments and Suggestions for Authors

Human OAT1, OAT3, OAT4 and OATP1A2 Facilitate the Renal Accumulation of Ochratoxin A

Mahadeo A, et al.,

In this manuscript, the authors examined the transport activity of ochratoxin A by various human drug transporters. Since ochratoxin A is found in many foods contaminated by fungi and is known to cause chronic kidney disease, understanding the mechanisms responsible for its renal accumulation is an important issue for human health.

The authors investigated the transport kinetics of ochratoxin A by major human renal drug transporters and found that OAT1, OAT3, OAT4, OATP1A2, and BCRP are capable of transporting ochratoxin A. These findings are valuable for elucidating the mechanisms underlying ochratoxin A accumulation in renal cells.

1) NPT (SLC17) anion transporters were not examined in this study. NPT4 has been reported as an ochratoxin A transporter. Since members of the NPT family share similar drug specificity, it is possible that NPT1–3 also transport ochratoxin A.

2) Fig 1: Since the raw data on ochratoxin A transport is not presented, it is difficult to evaluate the figure. Providing data for ochratoxin A with and without each transporter inhibitor would help clarify the validity of this experiment.

3) The current data only demonstrate transcellular transport related to BCRP, without directly showing BCRP-mediated transport activity. As BCRP vesicles are commercially available, direct measurement of BCRP transport is feasible and would strengthen the experimental evidence.

4) Lines 176,,,, : The specific radioactivity of ochratoxin A is not defined. The terms "Hot" and "Cold" are ambiguous and should be replaced with quantitative units such as Bq/nmol to ensure clarity.

5) The definition of "efflux" appears to be ambiguous. If described as an "efflux ratio," it can be understood as referring to transcellular transport. However, when simply referred to as "efflux," as in line 401, it may be confused with extracellular excretion. Clarifying the terminology would help avoid misunderstanding.

6) The resolution of the figure is too low to read its contents, including symbols, numerical values, axes, and labels.

7) In Figure 2, the transcellular transport of BCRP does not show saturated kinetics. This point requires further discussion. Although the KM value of BCRP for ochratoxin A is not provided, the absence of a saturated transport may suggest that the observed transport is due to passive diffusion rather than active transport.

Given that no transcellular transport is observed in BCRP(-) cells, the measurements in Figures 1b and 2f likely reflect BCRP-mediated transport.

In this assay system, which transporter is responsible for the uptake of ochratoxin A from the basolateral side into the cells? If no transporter is present on the basolateral side, passive diffusion across the basolateral membrane may become the rate-limiting step, potentially explaining the lack of saturation in the transport curve. If that is the case, special caution is needed when interpreting Figure 4, as it may be evaluating the effect on diffusion from the basolateral side rather than the activity of BCRP itself. Use of BCRP inhibitors that reduce BCRP activity may help this problem.

8) In line 436, the citation is given as Tsuda et al., but could this be a mistake? Should it be Zhang et al. instead?

9) Line 517: What does "BCRP-mediated ochratoxin A uptake" refer to?

BCRP is known as an efflux transporter. Is this referring to transport into vesicles using a membrane vesicle assay? Or does it refer to uptake from the basolateral side into the cell in a transcellular transport assay system?

If it refers to the latter, is the observed increase in uptake truly BCRP-mediated?

Clarification is needed regarding the direction and mechanism of transport being described.

10) Lines 377 and 496: The rationale for testing antioxidant polyphenols is not clearly explained. Are there any studies suggesting a link between these compounds and nephrotoxicity, and a correlation with ochratoxin A?

11) The transport of ochratoxin A by OAT1–4 and BCRP has been previously reported. It would be helpful to organize and compare these existing findings with the new data presented in this study.

Reviewer 2 Report

Comments and Suggestions for Authors

In this manuscript the authors demonstrated active uptake of ochratoxin A by renal uptake transporters OAT1, OAT3, OAT4, and OATP1A2. With the exception of OATP1A2 which was shown for the first time as a transporter relevant for ochratoxin A uptake, these results were confirmatory. With regard to the role of renal efflux transporters, the authors could only confirm the ochratoxin A transporter by BCRP, not by MRP2 and MRP4. In general, the results are of high quality and well presented. However, the interpretation of the results needs some improvements

  • The authors suggest that inhibition of BCRP in intestine by dietary polyphenols might increase the exposure of ochratoxin A. Since the oral availability of ochratoxin A is very high (>90%), the efflux transporters seem to have very limited impact of the intestinal absorption. Consequently, inhibition of BCRP is unlikely to increase the exposure of ochratoxin A
  • The authors suggest that, due to very high plasma protein binding, glomerular filtration would have minimal contribution to the clearance of ochratoxin A. This is not necessarily true. The pharmacokinetics of ochratoxin A in human and rat indicates a very low total clearance. In the absence of other clearance mechanisms, even a very low glomerular filtration clearance of ochratoxin A could contribute significantly to the total clearance. A rough estimation using the data from reference 12 showed e.g. that the total clearance of ochratoxin A in female rats is roughly equal to the unbound glomerular filtration rate, indicating a high contribution of uGFR to total clearance.
  • The results presented here suggest transporter-mediated cellular disposition of ochratoxin A. However, link between the transporter activities and the nephrotoxicity of ochratoxin A is not as obvious as suggested by the authors. The references 11 and 12 (by the way, there seems to be some mistakes regarding the use of both references: reference 9 and 10 on line 66 and 68 obviously refer to references 11 and 12) suggest some accumulation of ochratoxin A in kidney cells, the extent of the accumulation is not well characterized because the lack of data on binding to the intracellular content of renal cells. Since the clearance of ochratoxin is very low, the extensive exposure of ochratoxin could be the mean reason for the toxicity. Active uptake into kidney cells could contribute in addition.

Round 2

Reviewer 1 Report

Comments and Suggestions for Authors

Comment 1:

PT4 is a drug efflux factor that excretes drugs concentrated within cells. Considering this, the high Km value of NPT4 is reasonable, and its role should not be underestimated. However, I agree with the authors' estimation that its contribution of NPT4 is likely small.

Comment2:

Supplementary Figure 2 is missing. Therefore, the accuracy of the raw data cannot be assessed.

Comments 3 and 7:

Although obtaining the Km value of BCRP is important to reach such a conclusion, I agree that this is outside the scope of this report.

Author Response

We thank the reviewer for their comments and feedback. Please see our responses below. 

Comment 1:

NPT4 is a drug efflux factor that excretes drugs concentrated within cells. Considering this, the high Km value of NPT4 is reasonable, and its role should not be underestimated. However, I agree with the authors' estimation that its contribution of NPT4 is likely small.

Response 1: 

Thank you for pointing this out. We agree that it is common and reasonable for efflux transporters to have higher Km values relative to uptake transporters, as seen in this case for NPT4 vs. OAT1/3. Since NPT4 carries such a high Km value (~800 µM) for OTA, its intrinsic transport clearance could be estimated by Vmax/Km. Given NPT4's Km value and the OAT1/3 Km values quantified in our study (~2 to 3 µM), as well as their expression ratios (determined by scRNA-seq) (Uhlén et al. Molecular & cellular proteomics : MCP vol. 4,12 (2005): 1920-32.) of 2.3 : 1 : 1.3 for OAT1 : OAT3 : NPT4, the turnover rate (kcat) of NPT4 will have to be more than 1000× of OAT1/3 to favor the efflux of OTA. This is likely not physiological. As noted by the authors who reported the Km value of NPT4 (Jutabha et al. Journal of pharmacological sciences vol. 116,4 (2011): 392-6.), NPT4-mediated efflux is inefficient and may constrain net secretion of OTA, which favors PTEC intracellular OTA accumulation. 

Accordingly, we spotted a mistake in our discussion regarding this part. We have made the appropriate edits in the manuscript (lines 566 to 574). 

Comment 2:

Supplementary Figure 2 is missing. Therefore, the accuracy of the raw data cannot be assessed.

Response 2: 

We apologize for the confusion. Our files were previously uploaded as a single zip file and the old supplemental materials were not updated. We have now uploaded our revised materials individually. Thank you for pointing this out. 

Comments 3 and 7:

Although obtaining the Km value of BCRP is important to reach such a conclusion, I agree that this is outside the scope of this report.

Response 3: 

Thank you for your comment. We have added a statement in the discussion stating that the true Km value of BCRP transport of OTA will require future experiments in other systems, such as BCRP-expressing membrane vesicles (lines 506, 509 to 511).

Reviewer 2 Report

Comments and Suggestions for Authors

The revised version addressed adequately all comments I'd made.

Author Response

Thank you for reviewing our manuscript and sharing your constructive and insightful feedback with us . 

Round 3

Reviewer 1 Report

Comments and Suggestions for Authors

All of my questions are adequately addressed.

The manuscript is acceptable for pharmaceutics.